# Noninvasive Mapping of Angiotensin Converting Enzyme-2 in Pigeons Using Micro Positron Emission Tomography

**DOI:** 10.3390/life12060793

**Published:** 2022-05-26

**Authors:** Zilei Wang, Ziyu Liu, Lanxin Yang, Jin Ding, Feng Wang, Teli Liu, Zhi Yang, Chao Wang, Hua Zhu, Youping Liu

**Affiliations:** 1Department of Biochemistry and Molecular Biology, School of Basic Medical Sciences, Southwest Medical University, Luzhou 646000, China; wzl17710581527@163.com; 2Key Laboratory of Carcinogenesis and Translational Research (Ministry of Education/Beijing), Key Laboratory for Research and Evaluation of Radiopharmaceuticals (National Medical Products Administration), Department of Nuclear Medicine, Peking University Cancer Hospital & Institute, Beijing 100142, China; liu15104888166@163.com (Z.L.); ylx0081@163.com (L.Y.); as110_007@163.com (J.D.); windtigerwf@163.com (F.W.); liuteli123321@163.com (T.L.); pekyz@163.com (Z.Y.); 3Department of Nuclear Medicine, The Affiliated Hospital of Inner Mongolia Medical University, Hohhot 010050, China; 4Department of Science and Education, Beijing Nuclear Industry Hospital, Beijing 100045, China

**Keywords:** pigeon, ^68^Ga-HZ20, molecular imaging, angiotensin-converting enzyme 2

## Abstract

The ACE2 receptor, as the potential entrance site of SARS-CoV-2-affected cells, plays a crucial role in spreading infection. The DX600 peptide is a competitive inhibitor of ACE2. We previously constructed the ^68^Ga-labeled DOTA-DX600 (also known as ^68^Ga-HZ20) peptide and confirmed its ACE2 binding ability both in vitro and in vivo. In this research, we aimed to investigate the noninvasive mapping of ACE2 expression in fowl using ^68^Ga-HZ20 micro-PET. We chose pigeons as an animal model and first studied the administration method of ^68^Ga-HZ20 by direct site injection or intravenous injection. Then, the dynamic micro-PET scan of ^68^Ga-HZ20 was conducted at 0–40 min. Additionally, ^18^F-FDG was used for comparison. Finally, the pigeons were sacrificed, and the main organs were collected for further immunoPET and IHC staining. Micro PET/CT imaging results showed that ^68^Ga-HZ20 uptake was distributed from the heart at the preliminary injection to the kidneys, liver, stomach, and lungs over time, where the highest uptake was observed in the kidneys (SUV_max_ = 6.95, 20 min) and lung (SUV_max_ = 1.11, 20 min). Immunohistochemical experiments were carried out on its main organs. Compared to the SUVmax data, the IHC results showed that ACE2 was highly expressed in both kidneys and intestines, and the optimal imaging time was determined to be 20 min after injection through correlation analysis. These results indicated that ^68^Ga-HZ20 is a potential target molecule for SARS-CoV-2 in fowl, which is worthy of promotion and further study.

## 1. Introduction

Severe acute respiratory syndrome coronavirus-2 (SARS-CoV-2), the chief culprit of COVID-19, severely endangers humans worldwide [1]. SARS-CoV-2 showed a higher risk of transmission than SARS-CoV, which caused the 2002 outbreak. COVID-19 is mainly characterized by asymptomatic upper respiratory tract infections or potentially fatal atypical pneumonia associated with acute respiratory distress syndrome (ARDS) [2], which can lead to death in individuals [3].

Interspecies transmission of the virus is considered the main cause of coronavirus epidemics. SARS-CoV-2 is highly genetically related to a bat coronavirus (bat CoV RaTG13) with 96% genomic nucleotide sequence homology, verifying the kinship of SARS-CoV-2 with bat CoV RaTG13 [3]. However, due to the lack of direct contact between bats and humans, it is unlikely that the virus was transmitted directly to humans by bats. Thus, there may be an intermediate host for the transmission of SARS-CoV-2 to humans. For example, the coronavirus SARS CoV, which led to the 2003 SARS epidemic, was also derived from bats [4], but it was transmitted to humans via a variety of intermediate hosts, including civets and raccoons [5].

However, collecting samples from epidemic areas for virus detection is time-consuming and laborious, and the samples can easily be contaminated. Angiotensin converting enzyme-2 (ACE2) has been confirmed to be the cellular receptor for SARS-CoV-2, and ACE2 exists in two forms: a membrane-bound form and a soluble form. SARS-CoV-2 is a 65–125 nm spherical enveloped virus consisting of spike (S), membrane (M), envelope (E), and nucleocapsid (N) structural proteins. Among these is the spike(s) protein, a 150 kDa transmembrane glycoprotein that protrudes from the viral surface.

The entry of SARS-CoV-2 into host cells is mediated by the binding of viral spike protein to the membrane-binding form [6]. Its S protein binding to ACE2 induces a conformational rearrangement that drives membrane fusion, aiding viral entry into the host [7]. Therefore, angiotensin converting enzyme-2 (ACE2) is identified as the functional receptor for SARS-CoV-2 lung infection [8]. Receptor recognition is an important factor in determining viral host range and cross-species infection. The expression of the ACE2 receptor can be used for rapid screening and can narrow the scope of the SARS-CoV-2 intermediate host. The susceptibility of different species can be distinguished by the difference in ACE2 among species [9].

The screening of SARS-CoV-2 intermediate hosts is significant for studies of virus transmission and disease control. Current approaches to find intermediate hosts mostly employ contrasting amino acid sequences and predict the ability of SARS-CoV-2 to utilize different ACE2s. It has been reported that susceptibility was analyzed by comparing the differences in ACE2 protein sequences among species [9]. However, whole amino acid sequence recognition cannot accurately predict the utilization capacity of SARS-CoV-2. Nuclear medicine PET-CT, with suitable tracers as well as sensitive, accurate, and visual evaluation of receptor expression, has become an emerging examination method in the field.

In 2003, DX600, a novel ACE2-specific peptide inhibitor (Ki = 2.8 nm), was reported by Huang et al. [10]. DX600 is a high affinity ACE2 binding cyclic peptide with an intramolecular disulfide bond that was initially discovered from a phage display screen for ACE2 inhibition. Combined with nuclear medicine PET-CT, DX600 can accurately evaluate the susceptibility of COVID-5 by detecting the expression of the ACE2 receptor in vivo.

Although birds are not the natural host of the currently known coronaviruses, the classification of coronaviruses is based on genomic characteristics rather than host range. Therefore, we cannot rule out the possibility of new-type coronavirus infecting birds. It has been shown that the key site of the ACE2 receptor in pigeons, compared to other birds, has the highest coincidence with humans (75%) [11], making pigeons the most likely intermediate host. Therefore, we investigated the expression of the ACE2 receptor in pigeons to determine its role in the transmission of SARS-CoV-2.

^68^Ga (T_1/2_ = 68 min, β^+^ = 88%) is produced by the ^68^Ge/^68^Ga generator, which is independent of the cyclotron. It is suitable for radio probe labeling experiments in the laboratory. ACE2 is a carboxypeptidase formed by Ang I dissociating the C-terminal dipeptide of ACE from its C-terminus. DX600 (Ac-GDYSHCSPLRYYPWWKCTYPDPEGGG-NH2), which we selected, was not hydrolyzed by ACE2 and was specific to both competitive and noncompetitive inhibition of ACE2. DX600 has also been used as an ACE2 blocker in the treatment of SARS-CoV-2 [12]. 18F-FDG is one of the most widely used probes in the field of nuclear medicine [13]. A study has shown that lung lesions in patients with COVID-19 pneumonia are characterized by high uptake of 18F-FDG [14].

## 2. Materials and Methods

**Reagents, instruments and animals:** All chemicals, reagents and solvents are commercially purchased without further purification. DOTA-DX600 were custom synthesized by Shanghai QIANGYAO Biotechnology Company. Sep-Pak C18- Light cartridges were purchased from Waters (Milford, MA, USA). The product was analyzed by radio-high performance liquid chromatography (HPLC) (1200, Agilent, Santa Clara, CA, USA) equipped with γ detector (Flow-count, Bioscan, Washington, DC, USA), using a C18 column (Eclipse Plus C18, 4.5 × 250 mm, 5 µm, Agilent, Santa Clara, CA, USA). The PET/CT imaging studies of small animals were performed on the Mira PET/CT of PINGSENG Healthcare Inc. (Shanghai, China) and analyzed by Avator. Micro-PET/CT has a spatial resolution <1.0 mm and a sensitivity >12% (150–750 keV). Image acquisition for immunofluorescence staining was performed by Zeiss LSM780. Four adult female pigeons were purchased from Zhongning Pigeon Company.

**Radiolabeling and Quality Control of Radiopeptides**^68^Ga-HZ20: 195 µL 1.0 M NaOAc solution containing 50 µg DOTA-DX600 was added into 3.0 mL of ^68^GaCl_3_ freshly eluted from ^68^Ge–^68^Ga generator by 0.05 m of hydrochloric acid. The final pH of the reaction was controlled as 4.2 and the mixture was heated at 95 °C for 15 min. After the reaction was completed, the mixture was loaded onto an activated Sep-pak C18 cartridge. The cartridge was first washed with 5 mL of water to remove the free ^68^Ga, and then eluted with 0.5 mL 80% ethanol to obtain the product of ^68^Ga-HZ20. The solution was analyzed by radio-HPLC to assess the radiochemical purity. The HPLC was eluted with water-CH_3_CN system (Phase A: 0.1% TFA H_2_O; Phase B: 0.1% TFA CH_3_CN) using gradient elution (0–5 min 20% B; 5–10 min 20%–80% B; 10–12 min 80% B; 12–15 min 80%–20% B) at a flow rate of 1.0 mL/min.

**Injection route**: Chloral hydrate 10% was used for intramuscular anesthesia. 15 MBq of each route was taken for subcutaneous injection and pterygoid vein injection to compare the visualization difference of ^68^Ga-HZ20 between the two injection methods, and the pterygoid vein was finally chosen as the injection route.

**Micro PET/CT imaging**: Four healthy pigeons (female) were injected with 15 MBq ^68^Ga-HZ20 through the pterygoid vein, and then dynamic imaging was performed for 40 min to observe the dynamic distribution of the probe in the body. The images were acquired for 30 min, 40 min and 60 min, with a single PET scan time of 600 s. The Avator software was used to display the images, outline the region of interest for the main organs, record the maximum standard uptake values for the outlined areas, and calculate the dynamic changes in SUVmax for different organs. At the end of the 60 min imaging, the pigeons were euthanized and the main organs were obtained for in vitro imaging.

**Immunohistochemical experiment:** After organ imaging, ACE2 expression was studied by immunohistochemistry with a standard Envision technique in formalin-fixed, paraffin-embedded (FFPE) 4 µm specimens from pigeons. Tissue sections were dewaxed in xylene and rehydrated in gradient alcohol. Endogenous peroxidase was blocked with 3% dioxygen. Antigen retrieval was performed using a microwave by placing the tissue sections in citrate buffer (pH 6.0) and then cooling to room temperature. After closure in goat serum, the sections were incubated overnight in primary antibody against ACE2 (Abcam 108252) prepared at a dilution of 1:100 in a humidified chamber at −4 °C. The next day, after 30 min of rewarming at room temperature, the sections were stained using a Super sensitive™ polymer HRP IHC detection system (ZSGB-BIO PV-6000) according to the manufacturer’s instructions. A positive control was used with human normal kidney tissue, and a negative control was used with antibody diluent as the primary antibody. All specimens were examined by light microscopy. The IHC score of the FFPE samples was blindly quantified by two pathologists. We used a semiquantitative scoring system to assess the expression of this marker. The scoring was based on the intensity of immunoreactivity (negative—0; + −1; ++ −2; +++ −3) and tissue scanned by Aperio Versa 200, Leica.

**Immunofluorescent (IF) Staining**: Resuspend the cells after digestion, count, spread 5 × 104/500 μL into a twenty-four-well plate with small discs and adherent inside the cell culture incubator for 24 h; discard the medium, wash 3 times with pre-chilled PBS and add 4% paraformaldehyde, fix at room temperature for 15 min, 0.05% PBST wash 3 times; block cells at room temperature for 1 h using goat serum; then discard the blocking solution and dilute DX600-FITC (dilute to 5 μg/mL using blocking solution) Drop-add staining on small cell-coated discs, wash 5 times with 0.05% PBST after incubation at 4 °C for 2 h, then nucleus staining with DAPI (1:6000, blocking solution diluted) at room temperature for 10 min, 0.05% PBST wash 3 times and then wash once with PBS; finally observe and collect pictures under 63× oil microscope after sealing with 90% glycerol (10% PBS dilution).

**Western Blotting Assays**: cellular proteins were extracted in 40 mM Tris-HCl (pH 7.4) containing 150 mM NaCl and 1% (*v*/*v*) Triton X-100, supplemented with a cocktail of protease inhibitors and PMSF (nm). Equal amounts of protein were electrophoresed on a 8% SDS-PAGE gel and then transferred to NC membranes. After blocking with 5% skimmed milk, membranes were incubated, first with primary antibody (ab108252 1:1000) at 4°C overnight, then with HRP-conjugated goat anti-rabbit secondary antibody (Vector, Burlingame, CA, USA) for 1 hr at room temperature. After washing with 0.05% TBST, the blots were examined using the Super Enhanced Chemiluminescence Detection Kit (Applygen Technologies Inc., Beijing, China) and protein bands were visualized after exposing the membranes to Kodak X-ray film.

**Statistical Analysis**: the data were analyzed by GraphPad Prism 5 software and reported as mean ± SD. A *p* < 0.05 was considered statistically significant.

## 3. Results

Radiation labeling yield and radiochemical purity (RCP) > 98.0% of ^68^Ga-HZ20 were determined by radio high-performance liquid chromatography (Appendix A).

Intraperitoneal injection is a common method of administration, but because of the influence of the half-life of PET imaging agents, intravenous injection is usually more suitable to allow imaging agents to enter the systemic circulation quickly. Subcutaneous and pterygoid vein injection routes were taken, and static images were collected 40 min after injection (Figure 1A,B). After comparing the image results, the intravenous image better reflected the distribution of the probe in the pigeon, so pterygoid vein injection was selected. 

After the injection route was determined, the nonspecific radioactive probe FDG was used for PET imaging of pigeons. A healthy pigeon (female) was injected with 10 MBq ^18^F-FDG via the pterygoid vein, and static images were collected 40 min later (Figure 1C). MIP showed that the nonspecific radioactive probe FDG was only absorbed in the kidney, metabolized by the kidney and entered the cloaca. The above two groups of experiments verified the specific binding of ^68^Ga-HZ20 to ACE2.

We injected healthy pigeons with ^68^Ga-HZ20 for micro PET/CT scanning to evaluate the distribution of ACE2 in vivo. Each healthy pigeon (female) was injected with 15 MBq ^68^Ga-HZ20, and micro PET/CT images were dynamically collected. Images at 30 min (Figure 2E), 40 min (Figure 1B) and 60 min (Figure 3A) were collected after injection. The MIP diagram showed that within 0–40 min (Figure 2A–D) after injection, the drug entered the heart through the bloodstream and was exported, and the standard uptake value max (SUVmax) of the heart decreased from 3.08 to 0.84. At 20 min, kidney and lung uptake of ^68^Ga-HZ20 reached the highest level (SUVmax = 6.95, 1.11) and then showed a decreasing trend. At 40 min, liver uptake increased from 0.91 to 2.51, and stomach uptake also reached a peak (SUV_max_ = 0.63). Overall, since a few drugs were absorbed subcutaneously during injection, the absorption and metabolism were slow, the overall SUV value showed an upward trend, and at 40 min, some drugs were still circulating in the body and metabolized by the kidney.

After 60 min of injection, static images (Figure 3A) were collected and dissected. The main organs (Figure 3B) were taken for PET/CT imaging in vitro (Figure 3C). There was no significant change in cardiac uptake. Kidney uptake decreased to 3.05 with metabolism, while liver uptake decreased to 0.51. The specific uptake of stomach and lung was relatively stable compared with that at 40 min (SUV = 0.52; 0.41).

The dynamic changes of the probe in vivo were analyses (Figure 3D). We observed that probe uptake was mainly in the kidney at 20 min and then decreased over time, with uptake in lung tissue also peaking at 20 min.

Immunohistochemical results (Figure 4) showed that almost no ACE2 was expressed in the heart and stomach of pigeons. However, there were two ++ in the kidneys and the small intestine, which may provide a supporting basis for acute kidney injury and fecal–oral transmission of novel coronavirus. Moreover, (+) expression of ACE2 was also observed in the liver and lung. The above results were consistent with those of PET/CT. After a simple statistical analysis of the relationship between SUV peak 68Ga-HZ20 values and ACE2 immunohistochemical scores in different organs at the same time point, a certain positive correlation was found, for there was an obvious trend at 20 min (Figure 5), indicating that the suitable time of image acquisition is 20 min after radiotracer administration.

The results of cellular immunofluorescence showed that DX600-FITC was able to bind strongly and specifically to ACE2-positive cell membranes and slightly to the cell plasma, which is consistent with the histochemical localization of ACE2 as shown by immunohistochemical staining results, and again demonstrates that DX600 is able to specifically target ACE2. Additionally, as shown in Figure 6B,C, Western blot results verified the high expression of ACE2 protein on the HepG2-ACE2 cell line.

## 4. Discussion

ACE2 is the only pathway by which coronaviruses enter host cells. The expression level and specific distribution in different organs reflect the susceptibility, severity and prognosis of SARS-CoV-2 infection. ACE2, as a receptor of SARS-CoV-2, is reported to infect each other among humans, dogs, cats, tigers, minks and other animals (except snakes) [15]. This study conducted specific imaging of this target to explore the distribution of ACE2 in birds.

Compared with epidemiology results, the main organ of SARS-CoV-2 infection was the lung, which was also proven by lung uptake. However, due to the presence of alveoli and other cavities in the lung tissue, the probes showed low aggregation in the lung tissue and thus showed a low SUV value. The PET/CT results were confirmed by immunohistochemistry. This observation is consistent with the results of a recent highly sensitive RNA in situ mapping study, which showed that the expression of ACE2 gradually decreased from nose to respiratory tract and was consistent with the gradient through lung epithelial culture [16]. This gradient could be clearly seen in PET images of some volunteers in other studies by our team DX600 and HZ20 showed similar Kd of 98.7 and 100.0 nm, respectively, suggesting that nucleophile 68Ga has little or no effect on the affinity of the probe [17].

The ACE2 receptor is not only the transmission site of SARS-CoV-2 but also an important site of blood pressure regulation [18]. There is no doubt that the kidney and liver have the highest SUV and become the key organs of virus attack [19]. Kidney injury is common among SARS-CoV-2-infected people. In the process of curing, the attack of ACE2 by viruses leads to injury to organs with high expression [20]. It can be inferred that most of the complications and symptoms in infected individuals may manifest in the kidney and liver, except for the symptoms of the lung and respiratory tract.

The distribution of ACE2 receptors in the gastrointestinal tract suggests that the virus may infect pigeons through the fecal-oral route, and fecal contamination may be the medium of SARS-CoV-2 transmission in cities. Studies have found that 70% of SARS-CoV-2-infected people have diarrhea. A recent case reported that ACE2 was highly expressed in the esophagus and intestine [21]. The fecal–oral route of transmission has become more evident after a study wherein patients’ rectum swabs were positive for the virus even when their nasopharyngeal swabs were negative, making viral shedding from the digestive tract a potential route of transmission [22].

## 5. Conclusions

In summary, ^18^F-FDG, a conventional imaging agent in nuclear medicine, is not the characteristic imaging agent of ACE2. It has no specific uptake in healthy pigeons. It is not suitable for the evaluation of the SARS-CoV-2 virus receptor and cannot identify the key sites. ^68^Ga-HZ20, as an ACE2-specific binding probe with good radiochemical characteristics, combined with PET/CT technology provides a reliable and rapid technique for the tracing of target organs and hosts of SARS-CoV-2 and is a novel probe worthy of further development studies.

In this paper, the competitive ACE2 receptor inhibitor DX600 was selected, and a ^68^Ga-HZ20 radioactive probe was constructed by radiolabeling. Dynamic imaging was performed in pigeons to explore the in vivo pharmacokinetic characteristics and ACE2 receptor distribution and quantity and to infer the susceptibility sites and transmission routes of SARS-CoV-2 in birds. The results were in accordance with the clinical symptoms and microbial properties, which provided a convenient and accurate method to study the tracing of susceptible and infectious sources of different organisms.

## Figures and Tables

**Figure 1 life-12-00793-f001:**
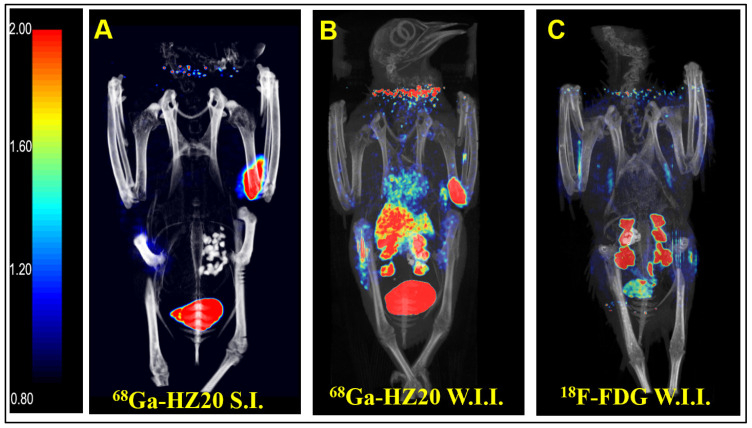
^68^Ga-HZ20 PET/CT images (**A**) 40 min after subcutaneous injection, (**B**) 40 min after Wing intravenous injection, and (**C**) ^18^F-FDG-Micro-PET/CT image 40 min after Wing intravenous injection.

**Figure 2 life-12-00793-f002:**
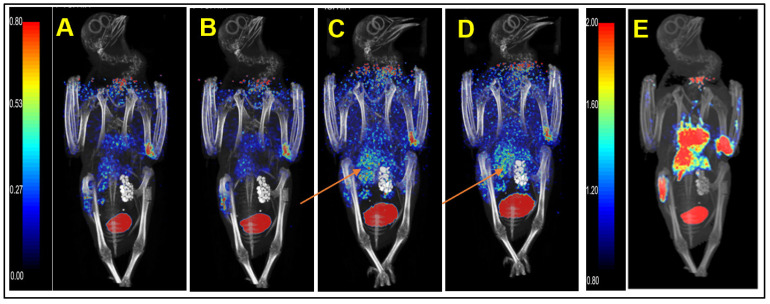
Micro-PET/CT imaging of ^68^Ga-HZ20 in pigeon at 10 min to 40 min (Slice) post injection (**A**) 10 min, (**B**) 20 min, (**C**) 30 min, and (**D**) 40 min, respectively. (**E**) Thirty minute imaging; the red arrows point to the kidney tissue of the pigeon.

**Figure 3 life-12-00793-f003:**
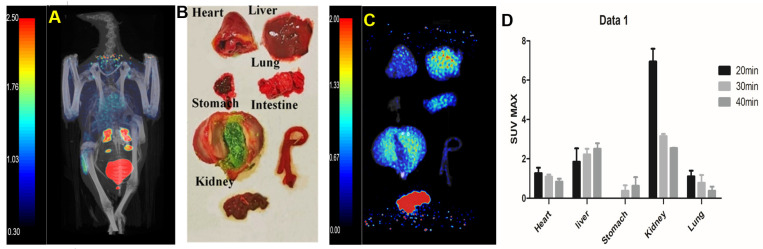
Micro-PET/CT imaging of ^68^Ga-HZ20 in pigeons at 60 min post-injection and organ imaging (**A**) 60 min, (**B**) isolated organ of pigeon, (**C**) PET images of isolated organs, and (**D**) biodistribution of ^68^Ga-HZ20 in pigeons at 20–40 min. The error bar was calculated as the standard deviation.

**Figure 4 life-12-00793-f004:**
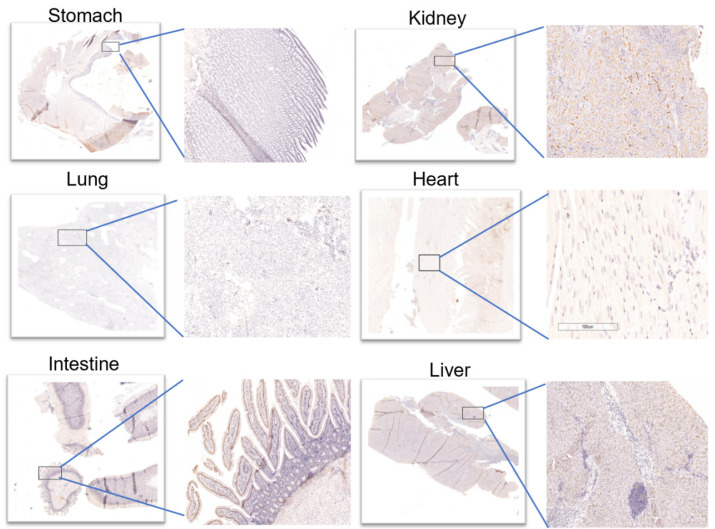
Images of IHC in different organs.

**Figure 5 life-12-00793-f005:**
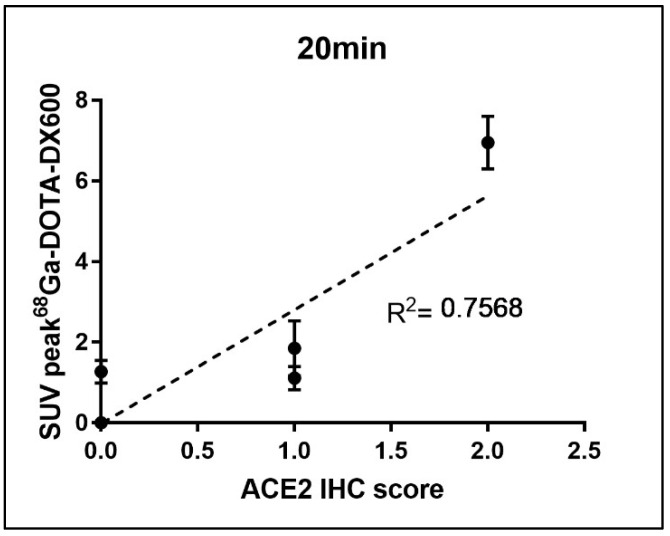
Relationship between ACE2 expression detected by immunohistochemistry and organ uptake (SUV peak) of ^68^Ga-HZ20.

**Figure 6 life-12-00793-f006:**
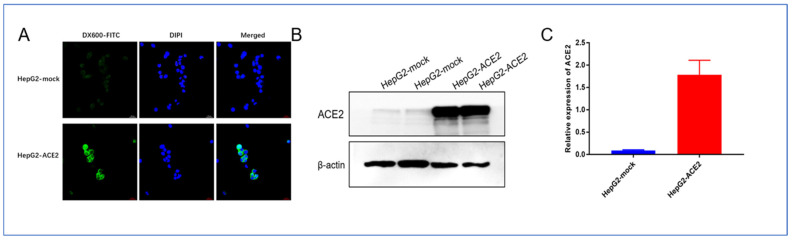
Cell level images. (**A**) Immunofluorescence staining of DX600-FITC on ACE2-positive and negative cells, (**B**) Expression of ACE2 protein on positive and negative cells, and (**C**) Western blot quantification of grey scale values of bands.

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
