# Peer review of "Noninvasive Mapping of Angiotensin Converting Enzyme-2 in Pigeons Using Micro Positron Emission Tomography"

_life, 2022, doi:10.3390/life12060793_

Round 1
Reviewer 1 Report
Comment for Author
In this manuscript, Wang et al. reported the construction of the 68Ga-labelled DOTA-DX600 PET imaging probe. 68Ga-DOTA-DX600 (also known as 68Ga-HZ20) peptide-based micro-PET can be used for the quantification of mammalian angiotensin-converting enzyme (ACE2) in vivo, specifically in this manuscript in pigeons. Immunohistochemical experiments were carried out on the pigeon’s main organs.
This research supposed to be the firstly construes direct correlations between functional imaging of ACE2 in pigeons and tissue level ACE2 protein expression (IHC) in vitro. After thoughtful analysis and work, the potential of 68Ga-HZ20 as a novel molecular targeting probe is demonstrated.
However, to improve the quality of professional publications, syntactic and text editing are also needed.
Overall, this is an experiment-based paper, fit for the research scope of “Life” Journal. Based on the above judgment, I recommended for publication of the manuscript in the special issue of “Life” Journal .
Here are some technical concerning for the further improvement of the manuscript .
- Please add cell experiments to prove the affinity of the probe.
- Please do additional cell tests to verify that the probe is in the same binding position at the cell and tissue levels.
- Please note the format of some references.
- Please note the caption of section 4 and 5 of the manuscript.
Here are some words/grammar flaws.
- Please note the writing of Angiotensin converting enzyme-2
- Please note the expression "a small number of drugs" on page 5
- A comprehensive proof reading is needed.
Author Response
Dear reviewer! we have completed our response to your comment, as described in the word document.

Reviewer 2 Report
This manuscript described a non-invasive method of mapping the ACE2 receptors in pigeons using a Ga-68 radiolabeled agent that targeted the ACE2 receptors. This method may be advantageous in finding the intermediate host of the SARS-CoV-2 virus. I recommend this manuscript to be accepted with moderate revision. Here are the specific comments:
- The authors must use a uniform font and font size throughout the manuscript.
- The description of radiolabeling and quality control section needs editing and appropriate terminology, superscript, and subscript must be used.
- The authors must provide a rationale for the use of subcutaneous injection method.
- It would be ideal to provide evidence of in vitro binding between the HZ20 peptide and the ACE2 receptors. A possible experiment for this purpose has been reported ( doi: 10.2967/jnumed.120.249748.)
- The experimental design lacked a negative control and a positive control. F-18 FDG was not appropriate for this purpose. It may have been better to use a Ga-68 DOTA containing peptide of similar size that did not bind to ACE2 receptors as a negative control in both in vitro and in vivo. The authors will need to prove: 1) the possible Ga-68 dissociation from the conjugate did not contribute to the biodistribution; 2) the uptakes of the tracer in ACE2 expression organs were ACE2-specific.
- The authors will need to explain the similar uptakes in stomach and lungs.
Author Response
Dear reviewer!we have completed our response to your comment, as described in the word document.

Reviewer 3 Report
Y.Liu et.al. studied the 68Ga- 29 HZ20 for noninvasive mapping of ACE2 in pigeons under m-PET. The article would have wide reader interest because of COVID-19 related research.
-author have to give reference of 18F-FDG for similar study in Pigeons/bird or give the explanation why selected the 18F-FDG for comparative study?
-A recent review article on 18F-FDG by O. Prante et al.,Pharamceuticals, 2021, 14, 1175 need to cite.
-comments on low uptake in stomach than liver (Fig.3. D) if SARS-CoV-2 transmission through fecal-oral route?
Author Response

(The authors gave the same response as above.)

Reviewer 4 Report
the authors present a manuscript on mapping of angiotensin-converting enzyme2 in pigeons using 68Ga-HZ20 micro-PET. Eventhough it is interesting to see noninvasive pidgeons images of FDG and the researc tracer, this paper may be rejected.
The article has several flaws and the authours present data but failed to turn them into relevant results. For example Figure 5: to present 4 data points (with zero as the 5.) for a regression calculation is discouraged in every statistics textbook. It is unclear why 4 animals were proposed for this experiment in the first place and an ethical clearance should not been given.
Given the rather short physiological half life of peptides in vivo the use of peptides in connection with viral infection imaging is debatable as a concept even if the ligand is known to show ACE 2 affinity.
In my opinion this work would has merit however the study design needs serious improvement and adequate animal numbers to address the conclusions made.
additional experiments needed, research not conducted correctly
Author Response

(The authors gave the same response as above.)

Reviewer 5 Report
This paper describes the comparison between FDG and 68Ga-HZ20 imaging with micro PET system using pegion.
The topic itself is interesting enough. I request minor revise to describe the experimental setup.
- What is the spatial resolution and sensitivity of micro PET you used. Please describe the information which is related to the understanding of accumulation.
- In PET image like Fig.1, 2, please indicate the size of image with bar.
- What is the dot-like accumulation in Figure 3. C at the bottom and top? It seems be also observed in Figure. 2.
- What is the error bar in Figure. 5?
Author Response

(The authors gave the same response as above.)

Round 2
Reviewer 4 Report
no comments
This manuscript is a resubmission of an earlier submission. The following is a list of the peer review reports and author responses from that submission.